# Ni-Infiltrated Spherical Porcelain Support as Potential Steam Reforming Microchannel Reactor

**DOI:** 10.3390/ma16041519

**Published:** 2023-02-11

**Authors:** Sandrine Ricote, William Grover Coors

**Affiliations:** 1Department of Mechanical Engineering, Colorado School of Mines, Golden, CO 80401, USA; 2Hydrogen Helix, Golden, CO 80401, USA

**Keywords:** porous ceramic, kaolinite, slip casting, ceramic fabrication, steam reforming, hydrogen production

## Abstract

This paper describes the fabrication of kaolinite (Al_2_O_3_-2SiO_2_-2H_2_O) spherical bulbs by slip casting. The bisque-fired parts present a porosity of about 30% with submicron porosity confirmed by scanning electron microscopy. In addition, plate-like grains with channels were observed. After nickel infiltration of the specimens, nanosized Ni particles covered the surfaces of the channels of these grains. Permeation tests in 5% H_2_ at 400 and 600 °C resulted in fluxes between 0.05 and 0.06 mol·m^−2^·s^−1^ at a pressure gradient of 200 MPa·m^−1^. Potential applications of these specimens include supports for hydrocarbon (namely ethanol) steam reforming.

## 1. Introduction

With the importance of hydrogen growing for the defossilization of the energy sector, renewable biofuels such as methanol [1], ethanol [2], and bio-oil [3] from cellulose are gaining prominence as replacements for fuels derived from petroleum. These fuels may be burned directly in heat engines, but a cleaner and more energy-efficient route is to extract the hydrogen from these hydrocarbons for use in fuel cells. The most common method is steam reforming, whereby the hydrocarbon molecules (in this example, ethanol) are decomposed into hydrogen, carbon monoxide, and carbon dioxide, in the presence of water at elevated temperatures on the catalysts deposited on high surface area substrates. Equation (1) gives the generalized reaction where 3 − n (3 ≤ n ≤ 1) is the mole fraction of carbon monoxide left in the reformate. CO is a catalyst poison for fuel cells and must be reduced to less than a few ppms. Steam reforming is an endothermic process, i.e., heat must be supplied in order to keep the process temperature steady. For the case of complete reforming, n = 3, and ∆H° = 347 kJ·mol^−1^. However, the heat required is only 58 kJ·mol^−1^ of hydrogen produced, which means making hydrogen by ethanol steam reforming (ESR) becomes very attractive compared with water electrolysis, which requires 248 kJ·mol^−1^ of hydrogen. Equation (1) is stoichiometric, but generally, coke formation must be prevented by increasing the steam-to-carbon (S/C) ratio [4].
C_2_H_6_O + n H_2_O ↔ (3 − n) CO + (n − 1) CO_2_ + (3 + n) H_2_, with 1 ≤ n ≤ 3(1)

Much progress has been made in the last twenty years with reforming catalysts supported on porous ceramic bodies and monoliths [4,5,6,7,8,9,10,11,12]. The most common catalysts are (1) Cu-based (Cu/ZnO on alumina, ceria, or zirconia), (2) Pd, Pt, or Ru based, and (3) Ni-based. While the Cu-based catalysts are very active and low cost, they suffer from low stability and pyrophoric nature [13]. Nickel catalysts are common candidates for ethanol steam reforming because of their high activity in C−C bond cleavage [14,15]. However, nickel suffers from coke formation and sintering. The latter can be inhibited if alloying with Ag or Cu [16,17]. Infiltration is a standard fabrication process to coat the pores of a substrate with the desired catalysts [5,10,11,16,17]: A solution containing the desired cations is impregnated in the porous supports as many times as needed to obtain the desired loading, and the sample is calcined in air to burn out all the organics then reduced to the metallic state.

Common support materials include alumina, magnesia, silica, ceria, and magnesium aluminate [11,18,19,20,21]. The challenge lies in optimizing the active area of the catalyst (increasing the porosity of the support) while retaining the mechanical strength of the oxide support. Microstructured reactors (composed of micrometer range channels) allow for process intensification due to improved heat/mass transfer and active volume [22]. A recent example from Divins et al. consists of a micromonolith made of two million regular channels of 3.3 μm in diameter [23].

A novel approach is described herein that uses the laminar structure of naturally occurring clay minerals as the structural element for depositing catalytically active nickel nanoparticles. The mineral, kaolinite, is particularly well-suited for this application. Kaolinite crystallite grains have cleavage planes that give the characteristic “slippery” texture in the water. These cleavage planes provide channels, about 100 nm wide, that form ideal microchannels for catalytic reactors and provide internal surface areas on the order of 10^6^ m^2^ per cubic meter. Highly porous and homogeneous membranes of kaolinite may be fabricated with ease from inexpensive and commonly distributed clay deposits around the planet. Such clays have formed the basis of pottery fabrication for ten millennia. In what follows, the fabrication of slip-cast porous ceramic bulbs and closed-end tubes infiltrated with nickel to form nanoparticles suitable for promoting the steam reforming and water gas shift reactions is described. This architecture has many benefits, including ease of manufacturing, scalability, and robustness in the steam reforming environment.

The novelty of this work lies in the combination of cheap and abundant material support (Al_2_O_3_-2SiO_2_-2H_2_O) with a unique microstructure (naturally present microchannels) and a widely used scalable fabrication technique (slip casting). Supports of various shapes were successfully fabricated. To the best of the authors’ knowledge, this manuscript is the first one to demonstrate the formation of Ni nanoparticles after infiltration of kaolin support.

## 2. Materials and Methods

Membrane bulbs and closed-end tubes were produced by traditional slip casting of porcelain slip and bisque firing to achieve the necessary porosity and microstructure. The bulbs were hollow with a wall thickness of about 1.3 mm. A 3 cm diameter bulb is shown in the cross-section in Figure 1a. The bulbs included a stem port for exchanging various gasses with the interior. The stem port was designed to interface with the balance of the system using a graphite compression fitting or glass seal. Closed-end tubes and other form factors were similarly fabricated.

The porous diffusion channels of the bulb were subsequently infiltrated with Ni nanoparticles, giving the bulbs a high surface area of catalytically active nickel for promoting the steam reforming reaction as hydrocarbon feed gas and steam passed through the wall of the bulb.

### 2.1. Slip

Porcelain slip made from 97% pure kaolin clay powder, Al_2_O_3_-2SiO_2_-2H_2_O, was purchased from Edgar Plastic Clay, FL, USA (median particle size of 1.36 microns). The slurry was prepared by mixing the kaolin clay powder and DI water in a 2:1 weight ratio, resulting in a highly viscous slurry. The viscosity was decreased by adding 1 wt.% of deflocculant (Darvan 7). It was found that the best-casted parts were obtained with a specific gravity between 1.6 and 1.7 g·cm^−2^. The parts were set to dry for 24–48 h at room temperature before being taken out of the molds.

### 2.2. Plaster Molds

Hollow spherical bulbs were slip cast as a single piece in a two-piece mold joined at the equator using techniques commonly employed in the pottery industry for making complex shapes. Precision molds were fabricated from a special plaster of Paris called “Perfect Cast”. The molds were made from mold master forms machined from Delrin plastic and polished. The diameter of the stem in the upper mold master is the only critical dimension. The stem was slightly tapered from 0.255″ at the base to 0.250″ at the tip to permit the mold master form to release from the plaster and also for ease of removal of the slip-cast parts from the mold. Closed-end tubes 1.25 cm in diameter and 9 cm long were fabricated in a similar fashion.

Figure 2 summarizes the steps of the slip-casting process for the bulb preparation described in Section 2.1 and Section 2.2.

### 2.3. Bisque Firing/Surface Preparation

The open porosity of the bulbs was obtained by firing them to a temperature sufficient for glassy phases to form but without the onset of vitrification. This process, bisque firing, results in practically no shrinkage. The bulbs were bisque-fired at 1100 °C for 1 h in air with heating and cooling rates of 2.8 °C·min^−1^. The resulting bulbs were very strong while exhibiting small pores and high porosity. Each fired bulb was pressure tested in water with 50 psi (0.34 MPa) air to detect any gross leaks due to cracks or pinholes.

### 2.4. Porosity and Bubble Point

The total open porosity was determined by weighing the amount of water taken up by the porcelain upon immersion. The fired density of the porcelain was determined to be 1.54 g·cm^−3^ from the geometry of a hemispherical section by diamond-sawing a bulb in half along the equator.

Bubble point testing was carried out on bulbs immersed in isopropanol to estimate the largest pore size at the surface. The principle behind bubble point testing is that the capillary force of liquids held in the pores of the ceramic must be overcome by applied pressure to force the liquid out of the porous network. Tiny bubbles begin to effervesce when the pressure applied to the inside of the bulb reaches the bubble point characteristic of the average microporosity at the surface. The process is described by the formula
d_avg_ = (4γcosθ)/∆P(2)
where d_avg_ is the pore diameter, and ∆P is the applied gauge pressure above ambient. γ is the surface tension of the liquid––21.7 dyne·cm^−1^ for isopropanol, and θ is the wetting angle–typically assumed to be zero.

### 2.5. Ni Infiltration and Reduction

A 1 molar Ni solution was prepared by dissolving the proper amount of nickel(II) acetate tetrahydrate (Alfa Aesar A13026.36) in distilled water. The infiltration was performed three times by immersing the specimen in the solution at room temperature followed by drying at 80 °C. The sample was finally calcined at 600 °C for 2 h in air to combust the organics of the acetate. This step was followed by the reduction of the residual nickel oxide in 50 mL·min^−1^ (sccm) 5% H_2_-balance Ar at 600 °C for 2 h. The amount of elemental nickel remaining in the specimen was determined by weighing to determine the difference between the specimen after Ni reduction and the specimen prior to infiltration. It was estimated on several samples to be between 0.04 and 0.05 g·cm^−3^. The metallic state of the nickel was confirmed by grinding a reduced bulb and observing the powder attracted to a magnet. Bulbs were dark in color after reduction due to light adsorption by the nanoparticles, which appeared to be uniform throughout the wall when sawed in half, as shown in Figure 1b.

### 2.6. Structural and Microstructural Characterization

Polished cross sections were obtained by polishing the sample down to a 1-micron finish. The samples were then rinsed in ethanol and ultrasonicated for 2 min. Back-scattered electron micrographs were collected with an Environmental Scanning Electron Microscope FEI QUANTA 600I (OR, USA). Fractured cross-sections of the porcelain bulb before and after infiltration were recorded with a JEOL JSM-7000 Field Emission Scanning Electron Microscope (FESEM, Japan). The cross-sections were gold coating to prevent charging.

X-ray diffractograms were obtained on powder samples (kaolin Edgar Plastic Clay before and after sintering) at CoorsTek using a Bruker D2 Phaser theta-theta diffractometer (copper tube 30 kV, 10 mA, dwell time 1 s, step size 0.02 degree).

### 2.7. Gas Permeation Testing

Bulb sealing into the test fixture was performed using standard stainless steel tube compression fittings with a graphite ferrule (GCFerrules no. 32272-50.) The graphite was malleable to prevent fracture of the brittle ceramic stem and enabled a compression seal to be fabricated that remained gas-tight in reducing atmosphere at temperatures up to 700 °C. The sphere, connected to a 1/4 inch gas inlet, was inserted into an MTI Corporation furnace and heated to 600 °C in 50 mL·min^−1^ (sccm) 5% H_2_-balance Ar. The flux through the bulb was measured using an Alicat mass flow controller. The fluxes were measured as a function of the inlet flux (through the stem of the bulb) over the range of pressure gradient divided by wall thickness (∆P/L) 50–300 MPa·m^−1^. Once the fluxes were collected at 600 °C, the sample was cooled to 400 °C, and similar measurements were performed. As-fired and Ni-infiltrated/reduced bulbs were tested according to this procedure.

## 3. Results and Discussion

### 3.1. Fired Bulbs

Since the porcelain body was made from liquid slip, the composition of the material was homogeneous throughout the membrane. Furthermore, slip casting ensured a uniform membrane wall thickness for the specimens produced. The wall thickness was controlled by the cast time between the introduction of the slip into the mold and when it was poured out, leaving a semirigid part to dry in the mold before extraction.

The casting time was 6 min for a wall thickness of 1.3 mm. Shrinkage during drying ensured that the part separated from the plaster mold and could be removed without damage in the green state. A sawed cross-section of a fired bulb is shown in Figure 1, illustrating the uniform wall thickness over the whole sample. As stated in Materials and Methods, the bisque firing step did not lead to further shrinking.

From the weight change of the bulb upon immersion in water, the porosity was estimated to be about 30%.

### 3.2. Phase Composition

The as-purchased Al_2_O_3_-2SiO_2_-2H_2_O (Edgar Plastic Clay) displayed the diffraction peaks for kaolinite (ICDD 01 078 2110) and some quartz (ICCD 00 046 1045) impurities. After sintering the part at 1100 °C, the material lost part of its crystallinity (the presence of amorphous phases responsible for the irregular baseline with noise), which is consistent with the study by Lee et al. [24]. As shown in Figure 3, the peaks for the sintered sample could be identified as quartz, aluminum oxide (ICDD 04 016 0538), and mullite (ICDD 00 015 0776).

### 3.3. Microstructure

#### 3.3.1. After Firing

Upon firing, the kaolinite underwent a phase change to microcrystalline metakaolinite, as obvious with the change in the XRD pattern and in agreement with the literature [24]. The micrographs of the polished cross-section of a bulb fired at 1100 °C, exemplified in Figure 4, show a submicron-sized, homogenously distributed porosity.

Further microstructural information was obtained by the bubble point determination, which was greater than 75 psi gauge pressure (0.52 MPa), meaning that the largest pores at the surface were less than about 100 nm.

High magnification secondary electron micrographs on a fractured cross-section, Figure 5, reveal the presence of several phases with various grain sizes (0.5 to over 10 microns) and shapes. One of noticeable importance is the plate-like microstructure highlighted in Figure 5b, with the porous channel being below 100 nm thick and several microns long. This feature is further zoomed in on Figure 6. It is noteworthy that the original mesostructure of the kaolinite grains is retained at the low firing temperature even though the intermediate metakaolinite phase between kaolinite and mullite is present. This is an essential feature of this clay-based process because the kaolinite structure is the result of the geologic process of weathering igneous rocks and cannot be fabricated artificially from alumina and silica.

#### 3.3.2. After Nickel Infiltration and Reduction

After the nickel infiltration, the spheres turned black (Figure 1b), and Ni nanoparticles were observed on the fractured cross-sections (Figure 7). Considering the weight gain during the infiltration process (0.04–0.05 g·cm^−3^), and assuming a particle size of 25 nm and its shape to be a hemisphere (V = 2/3·π·r^3^, with r the radius), the estimated density of Ni nanoparticles is on the order of 10^14^ per cubic centimeter. The Ni represents less than 0.5 vol.% and therefore cannot be detected using X-ray diffraction.

### 3.4. Gas Permeation

The flux of gas passing through the wall of the bulb per square centimeter determines the hydrogen production, which is the figure of merit for the device. From Equation (1) it is seen that for n = 1––that is, for CO as the sole reaction product––4 moles of H_2_ are generated per mole of ethanol, and the product stream is 67% H_2_. At completion, n = 3, and only CO_2_ is produced, and then 6 moles of H_2_ are produced per mole of ethanol, and the product gas is 75% H_2_. The actual value falls between these limits as long as no light hydrocarbon molecules, such as methane, make it through. It is the pore size and size distribution that control the flux of gas through the wall. The path length (or tortuosity) controls the residence time of the hydrocarbon species in the membrane and thus the probability of encountering a Ni nanoparticle and reacting before exiting the wall. If the pore size is too large, the coverage of the pore surfaces with Ni may be insufficient or too removed from the mean free path of the reaction gas to be effective. If the pores and flow channels are too small, the gas flux may be small, and the area required for commercial viability too great. Therefore, careful measurement and control of porosity are critical parameters for these bulbs. Moreover, all else considered equal, the permeation flux depends on the reciprocal of the wall thickness. It is also important that the porcelain material be homogeneous and have uniform wall thickness so that easy paths are eliminated, and the diffusion path length is the same throughout.

The pore size and size distribution are mostly determined by sintering temperature, and 1100 °C was found to optimize the total open porosity at about 30%. This temperature is also the temperature at which the platelet morphology of the mullite phase begins to form, ensuring good mechanical strength without loss of porosity due to shrinkage from sintering. Bubble point testing showed that the largest pores at the surface were less than 100 nm. The wall thickness was determined by the casting time. For the porcelain slip used in these experiments, adjusted to a specific gravity of 1.6 g·cm^−3^, a 6 min cast time resulted in a wall thickness of 1.3 ± 0.1 mm. The permeation flux of the bulb with a wall thickness of 1.3 mm adjusted for bulb area with a test gas of 5% H_2_-balanced Ar is plotted in Figure 8. The vertical axis is the measured gas flux in units of mol·m^−2^·s^−1^. The horizontal axis is the pressure gradient divided by wall thickness, ∆P/L, in units of MPa·m^−1^. The gas permeation of a noninfiltrated bulb was measured at 400 °C and 600 °C. Then, a nickel-infiltrated bulb was tested at 600 °C. No plausible explanation for the slight increase in permeation of the infiltrated bulb can be offered at this time.

For a practical system, it is necessary to optimize residence time to obtain maximum conversion efficiency, which may be accomplished by adjusting the applied pressure and wall thickness to obtain the desired hydrogen yield and purity. From Figure 8, it is seen that a pressure gradient of 200 MPa·m^−1^ results in a flux of about 0.055 mol·m^−2^·s^−1^. The actual tortuosity factor is estimated to be 5, giving a mean diffusion path length of 6.5 mm with an average residence time of about 4.8 s. In terms of hydrogen production, with a volume fraction of 70%, at 250 kJ·mol^−1^ (LHV) for hydrogen, 9.6 kW·m^−2^ would be produced. Thus, more than 100 m^2^ of membrane surface per megawatt of hydrogen production would be required. It is clear why the scale-up of such technology will need very inexpensive membranes.

## 4. Conclusions

The fabrication of kaolin spherical bulbs has been described, and 30% porous specimens with wall thicknesses of about 1.3 mm were obtained after bisque firing at 1100 °C for 1 h. Plate-like kaolinite grains with porous channels below 100 nm thick and several microns long were observed by scanning electron microscopy. The samples were infiltrated with a Ni-containing solution, and nanoparticles (about 25 nm) were observed within these channels.

Permeation studies in 5% H_2_ balance Ar were performed on a bulb before and after the infiltration process at 400 and 600 °C and resulted in fluxes between 0.05 and 0.06 mol·m^−2^·s^−1^ at a pressure gradient of 200 MPa·m^−1^. The usefulness of these bulbs ultimately hinges on demonstrating permeation of hydrogen at sufficient flux and purity under the operating conditions required by the membrane reactor environment.

## Figures and Tables

**Figure 1 materials-16-01519-f001:**
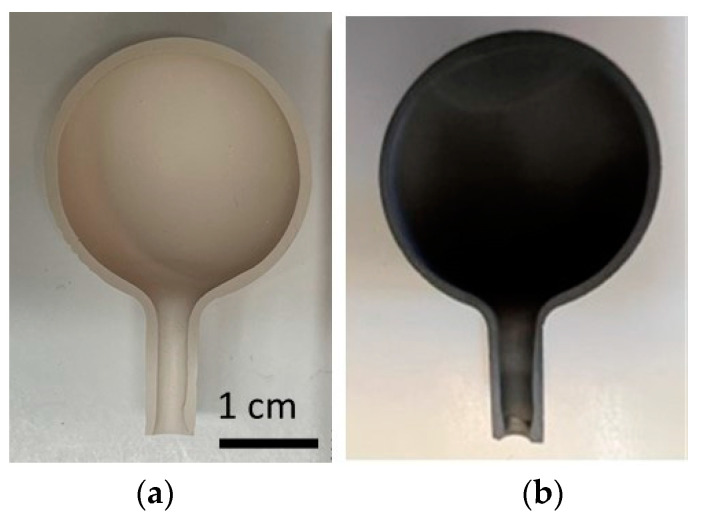
Hollow bulb in cross-section (**a**) before and (**b**) after the Ni infiltration and reduction. Both images are at the same scale.

**Figure 2 materials-16-01519-f002:**
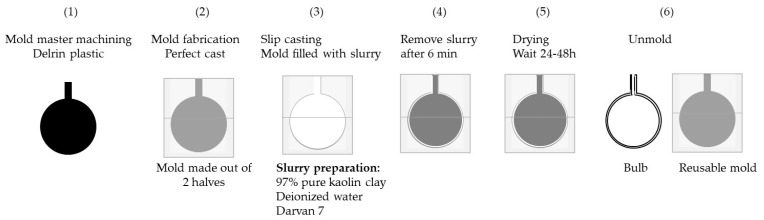
Schematic of the various steps for the slip casting of the bulbs of this study: (1) mold master machining, (2) mold fabrication, (3) slip casting, (4) removal of the slurry, (5) drying step, and (6) mold release.

**Figure 3 materials-16-01519-f003:**
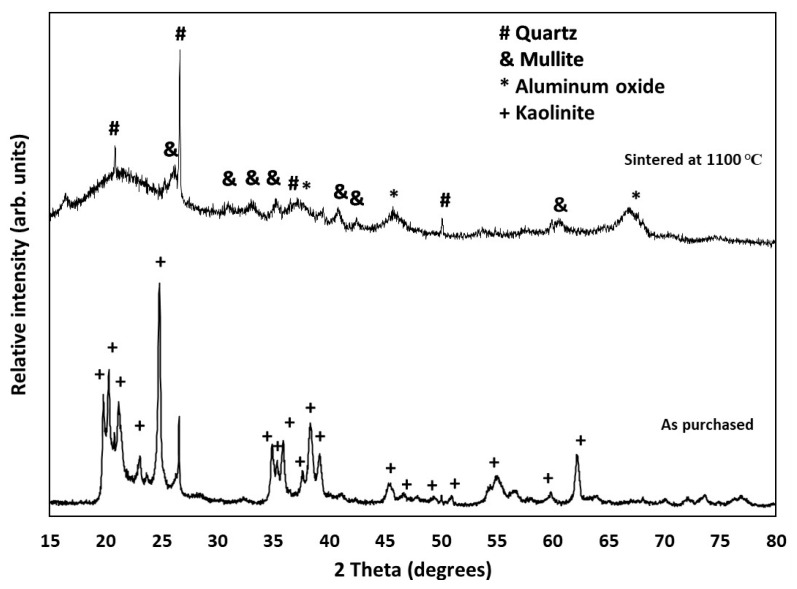
XRD pattern on the kaolinite powder as-purchased and after sintering at 1100 °C. The ICDD files used for the peak assignment are kaolinite (ICDD 01 078 2110), quartz (ICCD 00 046 1045), mullite (ICDD 00 015 0776), and Al_3_O_3.76_ (04 016 0538).

**Figure 4 materials-16-01519-f004:**
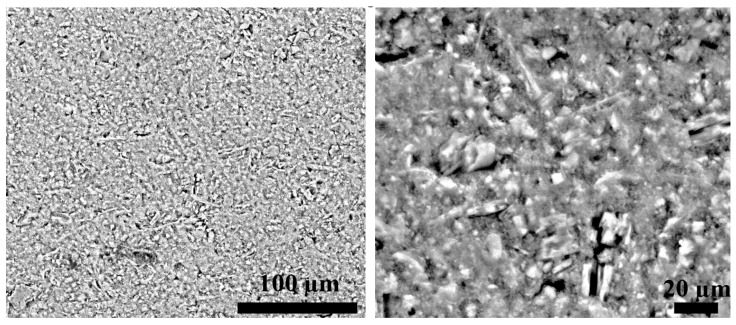
Back-scattered electron micrographs of a polished cross-section of a calcined bulb.

**Figure 5 materials-16-01519-f005:**
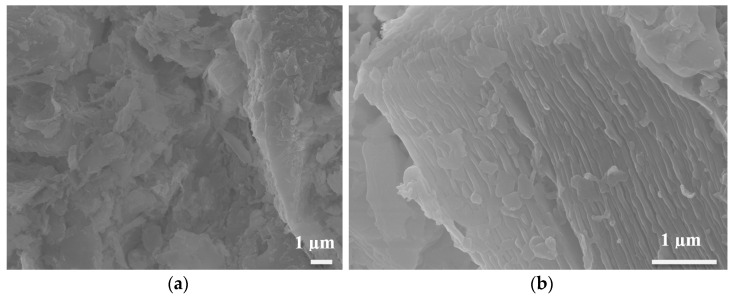
Secondary electron micrographs of a fractured cross-section of the calcined bulb. (**a**) Low magnification and (**b**) high magnification.

**Figure 6 materials-16-01519-f006:**
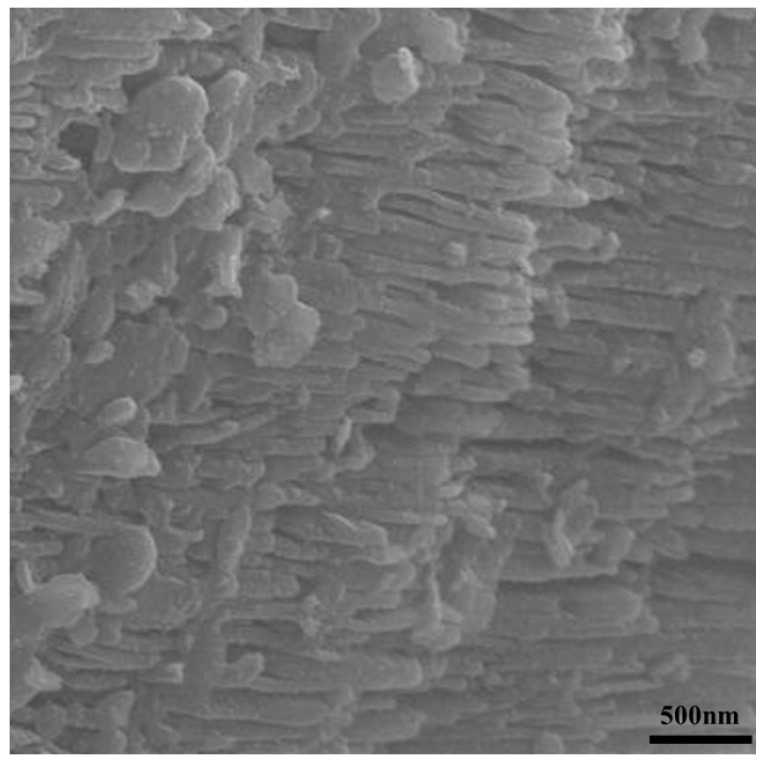
Secondary electron micrographs of a fractured cross-section of the calcined bulb—zoom on the plate-like grains.

**Figure 7 materials-16-01519-f007:**
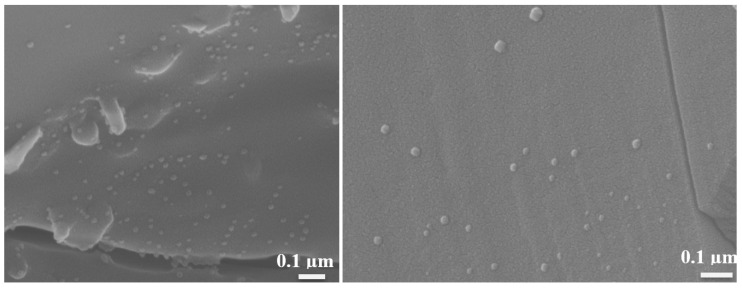
Secondary electron micrographs of a fractured cross-section of the calcined bulb after Ni infiltration and reduction.

**Figure 8 materials-16-01519-f008:**
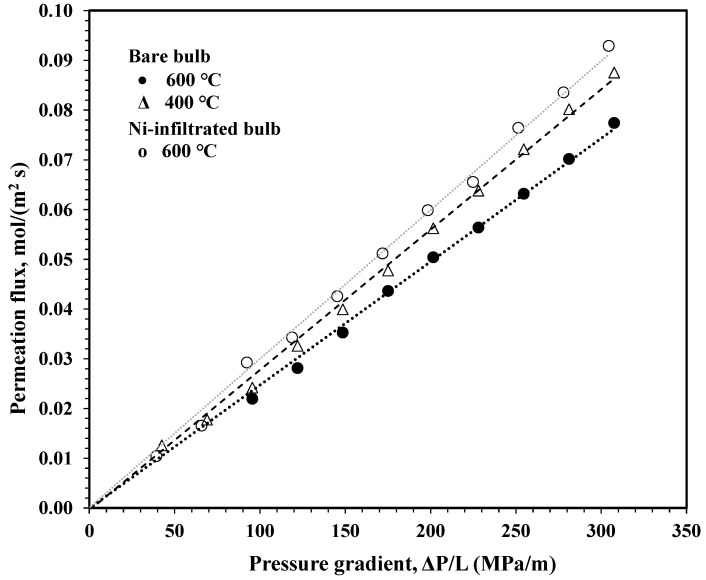
Gas permeation flux of 5% H_2_-bal Ar at 400 and 600 °C in units of mol·m^−2^·s^−1^ vs. pressure gradient, ∆P/L in units of MPa·m^−1^ for a bulb with 1.3 mm wall thickness, before and after infiltration.

## Data Availability

Original data are available upon request.

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
