# Peer review of "Ni-Infiltrated Spherical Porcelain Support as Potential Steam Reforming Microchannel Reactor"

_materials, 2023, doi:10.3390/ma16041519_

Round 1

Reviewer 1 Report

This article describes the fabrication of Ni-infiltrated kaolinite spherical bulbs by slip-casting for application as a steam-reforming microchannel reactor. The kaolinite raw material was determined by the XRD method before and after sintering at 1100 °C, the microstructure was examined by SEM, and the gas permeation was determined.

One question arose, that needs a minor revision:

1.      Please determine the total amount of Ni contained in such a bulb after infiltration, e.g. by the XRD method.

Author Response

We thank the reviewer for the comment.

The authors have added the weight of added nickel during the infiltration process= 0.04 to 0.05 g/cm3.

It corresponds to about 0.5 vol.%, which is below the detection limit of XRD.

In addition the results section has been reorganized and is more detailed. 

Reviewer 2 Report

The authors have conducted a study of the fabrication of kaolinite (Al2O3-2SiO2-2H2O) spherical bulbs by slip-casting and its composition and surface morphology. The processing technologie was discussed. However, the purpose and necessity of this study are not fully defined. For example,

1) The purpose and motivation of this study are not clearly discussed in the Introduction part;

2) For the Material and Method, the details about the materials are not clarified as well as the preparation method;

3) The characterization part is missing;

4) Please add the scale bar in Fig. 1 so that the readers can compare the dimension change after the Ni infiltration;

5) The pore size distribution details can be uncovered in section 3.1;

6) Sections 3.1 and 3.2 are not deeply discussed and only brief descriptions about Figs. 3-5 are not enough.

Author Response

The authors thank the reviewer for the comments. Below are the responses and the changes are highlighted in yellow in the revised version of the manuscript.

(1) A new paragraph about novelty has been added at the end of the introduction.

(2) Further details and a figure have been added in Materials and methods.

(3) The characterization part (X-ray diffraction, microscopy, flux measurement) is present in Materials and method.

(4) There is no change in size during the Ni infiltration process. We replaced figure 2a with a new figure with the same wall thickness as the infiltrated specimen. A scale bar has been added and it is specified in the caption that both figures are at the same scale.

(5) Two methods were used for the pore size measurements: SEM on a polished cross section and bubble point testing for the surface porosity. The porosity on the surface is significantly smaller than the one determined for the surface. Details have been added in the result part.

(6) The micrograph discussion had been improved. The important vitrification process is highlighted.

In addition the results section has been reorganized for more clarity.

Reviewer 3 Report

In this paper, Ni-infiltrated spherical porcelain support as potential steam reforming microchannel reactor was investigated. Some points remain unclear. I recommend the publication of this manuscript after major revision. 

1) Line 37, pay attention to the subscript of the chemical formula.

2) Line 60, “nanometers” should be replaced by “nm”.

3) Line 62, what does the unit “m2∙m-3” mean?

4) In the section of “Introduction”, the novelty and motivation of the work needs to be described further.

5) In section 2.1, how about the particle size of kaolin clay powder?

6) Line 114, psi should be converted to MPa.

7) Line 141, delete “were”.

8) In section 2.7, what’s the test standard followed?

9) In section 3.1, how about the pore size distribution?

10) It’s better to revise the subtitle of section 3.2 as phase composition.

11) Line 209, what does “the material loses part of its crystallinity” mean?

12) Please check the phase of Al3O3.76 in Fig. 3.

13) Line 232, how to estimate the density of Ni nanoparticles?

14) It’s better to add EDS pattern for identifying Ni nanoparticles.

15) Suggest put “section 3.1 Gas permeation” after section 3.3.

Author Response

The authors thank the reviewer for the comments. Below are the responses and the changes are highlighted in yellow in the revised version of the manuscript.

(1) And (2)- It has been corrected in the manuscript.

(3) The unit m2‧m-3 is used to report the area of the channels over the volume of the support. The higher the value, the higher the catalyst density. It was modified to m2 per cubic meter.

(4) A new paragraph about novelty has been added at the end of the introduction.

(5) The lay particle size has been added

(6) PSI were converted to MPa

(7) The authors do not think there is an error on line 141

(8) Details for the hydrogen permeation have been added.

(10) Title of section 3.2 has been changed.

(11) The sentence has been added: crystallinity (presence of amorphous phases responsible for the irregular base line with noise) page 5.

(12) Even though the most common phase for aluminum oxide is Al2O3, the phase Al3O3.76 fits better our patterns.

(13) This calculation is done by determining the number of Ni nanoparticles assuming that the Ni nanoparticles have a diameter of 25 nm and are shaped like half hemispheres. These details have been added to the manuscript.

(14) No EDX has been collected on the sample. It is confirmed that we formed Ni during the infiltration/reduction process has the crushed infiltration sample shows magnetic behavior.

In addition the results section has been reorganized and is more detailed.

Round 2

Reviewer 2 Report

The modification was accepted and the paper can be published now. 

Author Response

We thank the reviewer for accepting the revised version of our paper.

Reviewer 3 Report

1) It's better to replace ICDD file of Al3O3.76 with other alumina in Fig. 3.

2) What does "MTI" refer to in Line 159?

3)  In section 3.1, how about the pore size distribution?

Author Response

The authors thank the reviewer for looking at the revised version of the paper. The response to the comments are given below. And the changes have been highlighted in pink in the revised version.

1) 'It's better to replace ICDD file of Al3O3.76 with other alumina in Fig. 3.'

Better fitting results are obtained using the ICDD file for Al3O3.76 rather than that of Al2O3. Therefore the authors prefer keeping the actual ICDD file.

2) 'What does "MTI" refer to in Line 159?'

The name of the furnace company was corrected to MTI Corporation.

3)  'In section 3.1, how about the pore size distribution?'

The pore size distribution is generally determined using mercury porosimetry. We do not have access to one for this paper but we are planning to investigate this in our next paper related to reforming performance.